# Annotation of Potential Vaccine Targets and Designing of mRNA-Based Multi-Epitope Vaccine against Lumpy Skin Disease Virus via Reverse Vaccinology and Agent-Based Modeling

**DOI:** 10.3390/bioengineering10040430

**Published:** 2023-03-28

**Authors:** Sehrish Kakakhel, Abbas Ahmad, Wael A. Mahdi, Sultan Alshehri, Sara Aiman, Sara Begum, Sulaiman Shams, Mehnaz Kamal, Mohd. Imran, Faiyaz Shakeel, Asifullah Khan

**Affiliations:** 1Department of Biochemistry, Abdul Wali Khan University Mardan, Mardan 23200, KP, Pakistan; 2Department of Biotechnology, Abdul Wali Khan University Mardan, Mardan 23200, KP, Pakistan; 3Department of Pharmaceutics, College of Pharmacy, King Saud University, Riyadh 11451, Saudi Arabia; 4Department of Pharmaceutical Sciences, College of Pharmacy, AlMaarefa University, Ad Diriyah 13713, Saudi Arabia; 5Faculty of Environmental and Life Sciences, Beijing University of Technology, Beijing 100124, China; 6Department of Pharmaceutical Chemistry, College of Pharmacy, Prince Sattam Bin Abdulaziz University, Al-Kharj 11942, Saudi Arabia; 7Department of Pharmaceutical Chemistry, Faculty of Pharmacy, Northern Border University, Rafha 91911, Saudi Arabia

**Keywords:** lumpy skin disease virus, reverse vaccinology, vaccine designing, molecular modeling

## Abstract

Lumpy skin disease is a fatal emerging disease of cattle, which has started to gain extensive attention due to its rapid incursions across the globe. The disease epidemic causes economic loss and cattle morbidity. Currently, there are no specific treatments and safe vaccines against the lumpy skin disease virus (LSDV) to halt the spread of the disease. The current study uses genome-scan vaccinomics analyses to prioritize promiscuous vaccine candidate proteins of the LSDV. These proteins were subjected to top-ranked B- and T-cell epitope prediction based on their antigenicity, allergenicity, and toxicity values. The shortlisted epitopes were connected using appropriate linkers and adjuvant sequences to design multi-epitope vaccine constructs. Three vaccine constructs were prioritized based on their immunological and physicochemical properties. The model constructs were back-translated to nucleotide sequences and codons were optimized. The Kozak sequence with a start codon along with MITD, tPA, Goblin 5′, 3′ UTRs, and a poly(A) tail sequences were added to design a stable and highly immunogenic mRNA vaccine. Molecular docking followed by MD simulation analysis predicted significant binding affinity and stability of LSDV-V2 construct within bovine immune receptors and predicted it to be the top-ranked candidate to stimulate the humeral and cellular immunogenic responses. Furthermore, in silico restriction cloning predicted feasible gene expression of the LSDV-V2 construct in a bacterial expression vector. It could prove worthwhile to validate the predicted vaccine models experimentally and clinically against LSDV.

## 1. Introduction

Lumpy skin disease (LSD) is a transboundary bovine viral disease caused by the lumpy skin disease virus (LSDV). LSD is a serious threat to livestock breeding, substantially affects the livelihoods of farmers, and poses a constraint on the trade in livestock and their products [1]. The world health organization for animal health (WOAH, founded as OIE) has listed LDSV as a notifiable transboundary disease virus due to its rapid spread and significant economic impact. This disease has led to extensive economic losses in the livestock industry due to decreased milk production, infertility, abortions, mortality, restriction in cattle trading and movement, and the high costs of vaccination, treatment of infected cattle, and diagnostic services. LSDV is a linear double-stranded DNA virus that belongs to the genus capripoxvirus and family poxvirida. It is genetically similar to the other two species of capripox, goat pox and sheep pox. The genome size of LSDV is comparatively large (230–260 nm), containing 156 putative genes [2]. The disease is characterized by high fever, enlarged lymph nodes, emaciation, mucous in the mouth, reduced milk production, agalactia, nodular lesions, and infertility [3]. LSDV is primarily transmitted through arthropod vectors (mosquitoes, ticks, and biting flies), contaminated food and water, and nasal secretion. LSD outbreaks are more perceptible in the summer season coinciding with arthropods’ propagation in virus spreading. The severity of LSD often depends upon the animal breed, immune system, and age [2].

The OIE reported the prevalence of LSD in almost all countries in Africa, Asia, and Europe. LSD made its first appearance in Zambia in 1929 and then spread across the entire African continent [4], followed by cases emerging in Europe and Asia. Since 2019, the disease has spread to some major cattle-producing and trading countries across Asia and cases were reported in Bangladesh [5], India, China, Nepal, Sri Lanka, Bhutan, and Pakistan [6]. Due to limited access to an effective and reliable vaccine, LSD seems to be expanding in non-endemic areas. Live attenuated LSDV and ShPV have been successfully used as LSD vaccines in enzootic and outbreak areas. However, the use of live attenuated and homologous vaccines in the most severe scenario could lead to the risk of adverse reactions after vaccination. In the case of immunocompromised cattle, there is a chance of revival with live attenuated vaccines. At the time of an outbreak, vaccination may increase the risk of a recombination event between the field and vaccine strain in infected animals [7]. Because of these limitations and risks associated with live attenuated vaccines, a safe and reliable subunit-based vaccine is indispensable to halt and prevent the spread of LSDV. To date, there are no subunit vaccines available to fight against LSDV infection [8]. Multi-epitope-based vaccines are considered to be safer and more cost-effective than live attenuated vaccines [9].

The current study was planned to design an efficient and thermo-stable multi-epitope-based novel mRNA vaccine model by employing immunoinformatic and reverse vaccinology approaches. Reverse vaccinology facilitates vaccine model generation using genomic information and recursive computational analysis [10]. The designed vaccine should be capable of provoking the activation of CTL and HTL immune responses. The whole proteome of LSDV was used in the current study to decipher the potential B and T cell epitopes for the construction of a peptide multi-epitope vaccine. The efficiency of the modeled vaccine was assessed via its immunological and physiochemical properties. Additionally, molecular docking and MD simulation analyses were carried out to scrutinize the vaccine interaction within cattle TLR immune receptors. Several promising vaccine construct models were designed, and their experimental validation will be worthwhile for effective vaccine generation against LSDV.

## 2. Materials and Methods

The systematic workflow of the present study is depicted in Figure 1.

### 2.1. Protein Sequence Retrieval

The complete proteome of the LSDV Neethling 2490 strain was retrieved from the virus pathogen resource (ViPR) database [11]. CD-HIT analysis was performed to remove redundancy with a threshold of 90% sequence similarity to obtain non-paralogous sequences [12]. The resultant proteins were subjected to NCBI BLASTp against bovine proteins using a threshold E-value ≥ 10^−4^, a percent sequence identity of less than 50%, and a query coverage of less than 70% [13]. The non-paralogous viral-specific proteins were further examined and prioritized based on allergenicity, antigenicity, and toxicity. Highly antigenic, non-allergenic, and non-toxic proteins were selected for epitope prediction [14].

### 2.2. Epitope Prediction

#### 2.2.1. B-Cell Epitope Prediction

The shortlisted proteins were subjected to immune epitope prediction. B-cell epitopes trigger humoral immunity, which has the potential to invade pathogens by mediating immunoglobulin response. The ABCpred server (http://crdd.osdd.net/raghava/abcpred, accessed on 9 November 2022) with a cut-off score of 0.51 was employed to forecast 16-mer B-cell epitopes. The ABCpred server works via the neural network method using four parameters, namely, specificity, sensitivity, accuracy, and positive predictive value [15].

#### 2.2.2. Cytotoxic T-lymphocyte (CTL) Cell Epitope Prediction

The presentation of antigens to CTL by major histocompatibility class-I (MHC-I) is the preliminary step in activating the host immune response and producing memory cells against diseases. The IEDB consensus algorithm (https://www.iedb.org/home_v3.php, accessed on 10 November 2022) was utilized to identify the MHC class-I binding epitopes using all available bovine leukocyte antigens (BoLA) (Appendix A) [16]. IEDB offers different bovine MHC-I binding alleles. The MHC-I epitopes need to be significantly immunogenic to stimulate CD8+ T lymphocytes. Therefore, epitopes following a stringent threshold value of IC_50_ < 100 nm were prioritized for vaccine design.

#### 2.2.3. Helper T-lymphocyte (HTL) Epitope Prediction

The MCH class II-restricted helper T leukocyte (HTL) epitopes are involved in the initiation of both cellular and humoral immune responses. HTL epitopes were predicted using the NetMHCIIpan2.1 server prediction method [17]. The best binding MHC-II peptides for the chosen BoLA alleles were sorted based on the strong binding affinity, lowest percentile rank, and high prediction scores.

### 2.3. Evaluation of Predicted Epitopes

The allergenicity of the prioritized B- and T-cell epitopes was predicted using AllerTop v2.0 [18] with default parameters. Additionally, the antigenicity of the aforementioned B- and T-cell epitopes was determined by the VaxiJen v 2.0 server with a probability of 0.4 to identify the most potent antigenic epitopes [19]. The ToxinPred2 server was employed to evaluate the toxicity of the epitopes [20].

### 2.4. Multi-Epitope Vaccine Construction

The different combinations of shortlisted CTL, HTL, and B-cell epitopes were conjugated using amino acid linkers to construct a chimeric LSDV vaccine model. The GPGPG, KK, and AAY linkers were used to combine the B-cell, HTL, and CTL epitopes, respectively [21]. Beta defensin adjuvant sequence (Uniprot ID: P02584) was linked at the N terminal of the vaccine construct via EAAAK linker to improve immunogenicity. In addition, the non-natural pan DR (PADRE) 13 amino acid epitope (AKVAAWTLKAAAC) that induces CD4+ T cells, was also added to increase the potency and effectiveness of the vaccine. All these linkers were incorporated to achieve better expressions and bioactivity improvement and elicit a high immunogenic response. A highly immunogenic mRNA vaccine construct is reported to require a Kozak sequence, lead epitopes, a suitable adjuvant and linkers, and a stop codon [22]. The start codon must be a part of the Kozak sequence. The MHC-I targeting domain (MITD) sequence (UniProt ID: Q5S1P3) was incorporated into the 3′ region and tissue Plasminogen Activator (tPA) secretory signal sequence (UniProt ID: Q28198) was incorporated into the 5′ region of the vaccine construct [23]. The MITD sequence is required to direct CTL epitopes to the endoplasmic reticulum’s MHC-I compartment [24]. A 5′ cap, a poly(A) tail of 120–150 bases, and the globin 3′ and 5′ Untranslated Regions (UTRs) are essential for construct stability and were therefore added to the mRNA vaccine models [25].

### 2.5. Immunological and Physiochemical Properties Prediction

The purpose of vaccination is to elicit an immunological response against the disease. Therefore, the vaccine must be stable, non-allergic, antigenic, non-toxic, and possess good solubility. Hence, the allergenicity of the engineered multi-epitope peptide vaccine constructs was evaluated using AllerTop [26] and AlgPred v.2.0 [27]. The VaxiJen v2.0 server [19] and ANTIGENPro tools were utilized to predict the antigenicity behavior of the chimeric vaccine constructs [28]. The solubility of the constructs was analyzed using the Protein-sol tool [29]. In addition, the toxicity of the multi-epitope models was predicted using the ToxinPred2 web server [20]. The ProtParam online tool was used to screen various physicochemical properties of the vaccine construct [30].

### 2.6. Secondary Structure Prediction

The secondary structure elements of the vaccine constructs are predicted using PSIPRED 4.0 [31] server based on position-specific scoring matrices (PSSM). This server predicts the transmembrane helices, transmembrane topology, and recognition of the fold and domain regions in the peptide sequence. The secondary structure of the mRNA vaccine construct was predicted using the RNAfold tool of the ViennaRNA Package 2.0 [32]. The tool predicts the centroid secondary structure and calculates the minimal free energy (MFE) of the structure based on McCaskill’s algorithm [33].

### 2.7. Tertiary Structure Modeling and Refinement

The three-dimensional (3D) structures of the designed vaccine constructs were predicted using the Robtta tool, which implements comparative modeling or a de novo structure determination method for reliable 3D structure prediction [34]. The Galaxy Refine server was employed to refine the 3D structure of predicted vaccines [35]. Quality validation of refined 3D structures was performed using the PROCHECK program in SAVESv6.0—Structure Validation Server (ucla.edu) and ProSA-web server [36].

### 2.8. Molecular Docking

To effectively elicit the host’s immune response, the vaccine must interact properly with the host’s immune receptors. Therefore, molecular docking was carried out to screen the binding affinity of the different vaccine construct models to the host immune receptors. The 3D structures of bovine TLR2 (ID = AF-Q95LA9-F1) and TLR4 (ID = AF-Q9GL65-F1) [37] were retrieved from the AlphaFold Protein Structure Database [38]. Subsequently, the vaccine construct was intensively docked with receptors using the ClusPro 2.0 server [39]. PDBsum was employed to analyze the interaction between the proposed vaccine and TLRs residues in the docked complex [40].

### 2.9. Molecular Dynamic Simulation

Molecular dynamic (MD) simulation analysis of the LSDV-V2-TLR4 complex was conducted using the iModserver (iMODS). The server ensures vaccine stability, energy minimization, and the physical motions of atoms and molecules [41].

### 2.10. Codon Adaptation and In Silico Cloning

The sequences of vaccine constructs (V2) were subjected to the Java Codon Adaptation tool (JCAT) for reverse translation and codon adaptation to examine the peptide expression of the designed vaccine [42]. The GC content and CAI (codon adaptation index) were computed to access the expression level of the cloned vaccine sequence. A CAI score of 1.0 is considered ideal, however, a score of >0.8 may be considered good. The optimum percentage of GC content should range from 30% to 70%. GC values outside of this range result in adverse effects on transcription and translation, which can be used to determine the degree of protein expression [43]. The snapgene tool (https://www.snapgene.com/, accessed on 4 January 2023) was used to clone the adapted sequence of the finalized vaccine construct in the *E. coli* expression system. The pET28a+ expression vector for this purpose was obtained from the addgene server (https://www.addgene.org/, accessed on 4 January 2023).

### 2.11. Immune Simulation

In silico immune simulation of the top-ranked prioritized engineered vaccine construct was conducted using the C-ImmSim server to elucidate the immunogenic potential of the vaccine. This server employs a position-specific score matrix (PSSM) along with several machine-learning techniques. It is a cellular-level agent-based model that collects information on humoral and cellular responses of the mammalian immune system in response to antigens. Immune simulation for the vaccine model was performed based on a protocol previously employed by Aiman et al., 2022 [13]. The simulation parameters were set as default for a period of 1 h, 84 h, and 168 h along with selected leukocyte antigens, i.e., HLA-A*0101 and A*0201, HLA-B*0702 and B*0702 and B*3901, and HLA-DRB1*0101 and DRB1*0401. The volume of the simulation was set at 10 and immune simulation analysis was conducted based on 1000 simulation steps.

## 3. Results

### 3.1. LSDV Vaccine Candidate Proteins Prediction

The entire proteome of LSDV Neethling 2490 strain was obtained from VIPR in FASTA format (Appendix A). After removing redundancy via CD-hit suit, a total of 123 non-paralogous proteins were acquired (Appendix A). The bovine non-homologous protein sequences of the LSDV were further scanned based on allergenicity, antigenicity, and toxicity. Highly antigenic, non-allergic, and non-toxic proteins were shortlisted for epitope prediction (Table 1). 

### 3.2. B-Cell Epitopes Prediction

The shortlisted proteins were submitted to the ABCpred server to predict the lead B-cell epitopes. Nine potential epitopes were selected based on prediction score, toxicity, allergenicity, and antigenicity. A threshold value of 0.7 was used to predict B-cell epitopes for vaccine design. The top prioritized B-cell epitopes along with prediction score, binding, allergenicity, toxicity, and antigenicity scores are enlisted in Table 2.

### 3.3. Prediction of CTL and HTL Epitopes

Subsequently, the shortlisted vaccine candidate proteins were subjected to the IEDB and NetMHCIIpan web servers for MHC1 and MHCII epitope prediction, respectively, using bovine alleles. T-cell epitopes (CTL and HTL) were scrutinized using IC_50_ value less than 100 nM, lowest percentile rank, and strong binding affinity with different BoLA alleles. Finally, nine epitopes were prioritized for downstream analysis based on their antigenicity, allergenicity, and toxicity (Table 3).

### 3.4. Chimeric Vaccine Construct

The top-ranked epitopes were combined in different combinations along with adjuvants and linkers sequences to design multi-epitope vaccine constructs. Beta-defensin adjuvant was fused at the N-terminal of the designed vaccine construct to boost the host immune response. Different linkers (AAY, GPGPGC, etc.) were added to increase the stability and immunogenicity of the chimeric vaccine constructs (see methodology section). The linkers stabilize the conformation of the modeled vaccine constructs [44]. Finally, three different multi-epitope vaccine constructs with 353 amino acid residues were generated (Appendix A). The designed vaccine models were screened for the presence of allergenicity, antigenicity, and solubility potentials using multiple tools. The antigenicity of the vaccine models was evaluated using VaxiJen v2.0 and the ANTIGENpro server, with a threshold score of >0.5, which indicates a significant antigenic nature of the chimeric vaccine constructs (Appendix A). The ANTIGENpro assesses the antigenicity by 10-fold cross-validating proteins against known datasets and looking for the protective aspects of antigenic sequences [28]. AlgPred and Allertop predicted the vaccine models as non-allergenic using a cutoff value ≤ −0.4. The SoLpred server predicted the modeled vaccine constructs that exhibit the best solubility. The allergenicity, solubility, and antigenicity scores of the three top-ranked vaccine models are given in Appendix A.

### 3.5. Physiochemical Parameters Prediction of Vaccine

The various physiochemical properties of the modeled vaccines were evaluated via ProtParam (Table 4). The molecular weight of the three vaccines was observed to be around 30 kDa. A protein molecular weight below 110 kDa is a good choice for large-scale production owing to the ease of purification. The theoretical PI ranges from 9.67 to 9.64, which indicates the basic nature of the proposed vaccine constructs. The estimated high aliphatic index score ranging from 77.11 to 79.08 shows the thermo-stability of the chimeric vaccine construct. The predicted GRAVY (hydropathicity) ranges from −0.200 to −0.061, representing the hydrophilic nature of vaccine constructs. Likewise, the low instability index value indicates the vaccine construct stability. All of the designed vaccines were predicted to have a suitable half-life of 30 h.

### 3.6. Secondary Structure Prediction

The predicted secondary structures of the peptide vaccine constructs showed that all the designed LSDV constructs were comprised of 39–42% α-helix, 21–28% extended strand, 8–9% β-turns, and 26–30% random coils in their structures (Table 5 and Appendix A). The secondary structure of the mRNA was predicted by the RNAFold server. The proposed mRNA structure models were stable with the minimum free energy (MFE) of −299.35 kcal/mol for LSDV-V1, −269.35 kcal/mol for LSDV-V2, and −287.09 kcal/mol for LSDV-V3 in a thermodynamic ensemble. The centroid secondary structures had a minimum free energy of −237.70 kcal/mol for LSDV-V1, −169.48 kcal/mol for LSDV-V2, and −172.68 kcal/mol for LSDV-V3 in dot-bracket notation (Figure 2 and Appendix A).

### 3.7. D structure Evaluation, Refining, and Validation

A stable and efficient 3D structure of a chimeric vaccine construct is essential to evaluate the molecular interactions with the host immune receptor protein [45]. Hence, the Robetta server was used to generate the 3D structure of the designed chimeric vaccine constructs (Figure 3A), followed by refinement with the GalaxyRefine server (Figure 3B). The GalaxyRefine server produced five refined models and the best model was prioritized based on a high score for GDT-HA and a low score for Rama-favored (Appendix A). The Ramachandran plot was used as a quality check to validate the refined 3D structure of the vaccine constructs (Figure 3C). The Ramachandran plot predicted that 94.5%, 88.2%, and 89.9% of the residues of the V1, V2, and V3 constructs were in the most favored regions, respectively (Appendix A). These values indicate the good quality of all the models. The Z-score computed by the ProSA-web server for the top prioritized construct LSDV-V2 was −8.25 (Figure 3D and Appendix A).

### 3.8. Molecular Docking of Vaccine with TLRs 

Understanding the interaction between the proposed vaccine and bovine TLR immune receptors is indispensable for the elucidation of an effective and stable immune response. The ClusPro tool was used to perform the docking interaction of all three vaccine constructs with TLR2 and TLR4 immune receptors. The LDSV-V2 construct returns the top-ranked lowest binding energy (i.e., −1157.4 kcal/mol) against the TLR4 receptor (Figure 4A). During the PPI interaction of the LSDV-V2 construct with TLR4, a total of 21 hydrogen bonds (Appendix A), 168 non-bonded contacts, and six salt-bridge interactions were observed (Figure 4B; Appendix A). 

### 3.9. MD Simulation

MD simulation analysis was performed to ascertain the stability of the LSDV-V2-TLR4 complex using the IMOD server. The main chain deformity graph of the LSDV-V2-TLR4 complex depicts peaks that show a deformed region of the vaccine (Figure 5A). The graph in Figure 5B illustrates the association between the NMA and PDB fields in the complex. The eigenvalue calculated for the vaccine TLR4 complex was 7.122024e−05 (Figure 5C). The variance graph is inversely related to the eigenvalue and is connected to each normal mode of the complex, representing individual (purple) and cumulative (green) variances (Figure 5D). The covariance matrix of the complex represents the coupling between different pairs of correlated (red), anti-correlated (blue), or uncorrelated (white) atomic motions in the complex molecule (Figure 5E). An elastic network model of LSDV-V2 was also generated (Figure 5F), which differentiates the atom pairs connected by springs. In the diagram, each dot represents a spring between the corresponding atom pairs and its color relates to the stiffness of the spring. The darker grey color represents that the springs were more rigid. Eventually, these results predicted the stability of the LSDV-V2-TLR4 complex. 

### 3.10. Codon Adaptation and In Silico Cloning

The expression capability of a vaccine construct is the key step in vaccine design. We examined the expression of the designed construct in *E. coli* K12 strains, i.e., having a distinctive expression system, which requires codon adaptation. The JCAT web server was used to ensure codon optimization inside the *E. coli* expression system (Appendix A). The observed optimized cDNA has a CAI value of 1.0 and a GC content of 49.77%, representing the favorable expression of the vaccine construct in the *E. coli* strain K12. The restriction enzymes salI and Xbal were introduced at the N and C termini of the vaccine construct, respectively (Figure 6), and the final optimized sequences were cloned into the pET28a (+) vector plasmid using SnapGene software. The final length of the recombinant plasmid was 6646 bp.

### 3.11. Immune Simulation

Immune simulation of the prioritized LSDV-V2 vaccine construct was performed to evaluate the cellular and humoral responses generated by the host immune system against the vaccine. The result of immune simulation predicted a substantial increase in the secondary responses induced by the prioritized engineered LSDV-V2 vaccine construct. This trend is theoretically consistent with the progression of the real-time immune response. An increased level of IgM was evident in the primary simulated immune response. The secondary and tertiary immune simulated responses exhibited a substantial increase in the B-cell populations, as well as a marked increase in IgG1+ IgG2, IgM, and IgM + IgG antibodies. A decrease in the antigen levels was also observed (Appendix A). This implies the development of immunological memory, which is supported by the elevated level of memory B-cell population and isotype switching. These results predicted a rapid antigen decrease after the chimeric antigen exposure (Appendix A). The cytotoxic (TC) and helper (TH) T-cell populations were noted to have an elevated response, with the development of respective memory upon subsequent antigen exposure (Appendix A). Furthermore, during the immunization phase macrophages, dendritic cells, and the natural killer cell population were predicted to elicit and maintain increased levels (Appendix A). Elevated concentrations of cytokines, such as IFN-y, and interleukins, i.e., IL-2, were also evident (Appendix A). These findings suggest that the predicted vaccine model LSDV-V2 triggered encouraging immune responses against LSDV.

## 4. Discussion

Lumpy skin disease is a transboundary and re-merging notifiable viral disease on the OIE list with significant economic loss and morbidity in livestock [46]. LSDV spreads rapidly and causes outbreaks in different areas. Therefore, there is an immediate need for protection strategies and therapeutics to provide defense against LSDV disease. Vaccination is the only strategy that can effectively halt and prevent the spread of lumpy skin disease alongside the culling of infected animals and movement restriction of livestock in endemic regions [2]. Nevertheless, at the moment of outbreak, the selection of the best vaccine is a major challenge for veterinary authorities and farmers. The innocuous and efficacious multi-subunit vaccines against LSDV may be successfully utilized to control the disease. Immunoinoformatics-based mRNA vaccines are more widely accepted and provide immunological memory for several years. The first successful instance of mRNA therapeutics was reported in 1990, and since then effective and versatile mRNA vaccines have been developed against HIV-1, Zika, rabies, influenza virus, and other viruses [47,48,49,50,51]. LSDV surface proteins were prioritized to predict potential T- and B-cell epitopes for designing a multi-peptide-based vaccine construct capable of eliciting a strong CD4+ and CD8+ T-cell-associated immune response [52]. Kar et al. (2022) predicted a vaccine model based on MHC-I and B-cell epitopes based on limited protein sequences of the LSDV. Likewise, another study recently predicted an immunoinformatics-based vaccine construct model [53]; however, the results and strategy of the latter study are misleading as the analysis was performed based on human host immune cell receptors even though LSDV is a bovine rather than human pathogen. The current study was therefore conducted to design a multi-epitope-based mRNA vaccine construct against LSDV that is capable to induce immunogenic responses in infected animals. This approach is cost effective, reliable, quick, and efficient for designing vaccine. The approach used in the current study has been validated experimentally in mice models for several pathogens [54,55,56]. 

The entire proteome set of the LSDV Neethling 2490 strain was analyzed to search for potential vaccine candidate targets. The B-cell, MHC-I, and MHC-II epitopes were predicted based on stringent criteria. The B-cells trigger a humoral immune response that neutralizes viruses and creates a memory to protect against any subsequent exposures, but this often has low effectiveness and diminishes over time [57]. Conversely, T-cells (CTL, HTL) evoke a cell-mediated immune response that restricts the spread of pathogens either by destroying infected cells or by secreting antiviral cytokines that promote lifelong immunity. [58]. Therefore, the current vaccine construct was designed from multiple B-cell, CTL, and HTL epitopes in combination with linkers and beta defensive as an adjuvant. An adjuvant increases immunogenicity, while AAY and GPGPG linkers enhance the stabilization and expression of vaccine constructs. The top-ranked vaccine models predicted in the current study were observed to be highly antigenic, non-allergenic, and non-toxic. The analysis of physiochemical properties suggested that the designed vaccine is stable, basic, and hydrophobic indicating its potential in stimulating a strong immunogenic response. The knowledge of the 3D structure of a vaccine is essential for understanding the antigen-receptor interaction. After a refining process, the 3D structures of the proposed LSDV vaccines significantly improved and displayed the desired stability as predicted by Ramachandran plot analysis. Previous research has shown that TLRs are involved in the identification of viral peptide structures, which are responsible for the activation of the immune response [59]. Hence, molecular docking analysis of the LSDV vaccines was followed against TLR2 and TLR4 receptors. The docking result predicted feasible biochemical interactions between vaccine constructs and TLRs, especially in the case of LSDV-V2 against TLR4. The thermodynamic stability of the LSDV-V2-TLR4 docked complex was verified by downstream MD simulation analysis as well. This suggests that the vaccine can activate TLRs, and possibly generate immunogenic responses. Based on immune simulation analysis, the designed vaccine construct is capable of eliciting the host immune system to generate cellular and humoral responses against the designed mRNA vaccine. We speculate that the mRNA vaccine dose will efficiently translate once it reaches the cytosol of a bovine host as the construct is based on bovine-specific tPA and MITD sequences. The current study provides the groundwork to predict a highly immunogenic vaccine with a potential to elicit strong immune responses in the bovine host immune system. Experimental follow-up and clinical assays are required to validate the findings of this study.

## 5. Conclusions

The current study used subtractive genomics and reverse vaccinology approaches to design a novel mRNA vaccine against LSDV. The lead B- and T-cell epitopes were anticipated from the shortlisted proteins and utilized in the designing of multi-epitope vaccine constructs. The lead construct model was found to hold all the desirable characteristics to possibly elicit a robust immune response with non-allergenicity. Molecular docking and MD simulation confirmed the strong binding affinity of the proposed vaccine construct with immune receptor proteins. Immune simulation analysis predicted feasible immune response generation against the lead LSDV-V2 construct. The model mRNA vaccine needs experimental validation to ensure its efficacy against LSDV infection. 

## Figures and Tables

**Figure 1 bioengineering-10-00430-f001:**
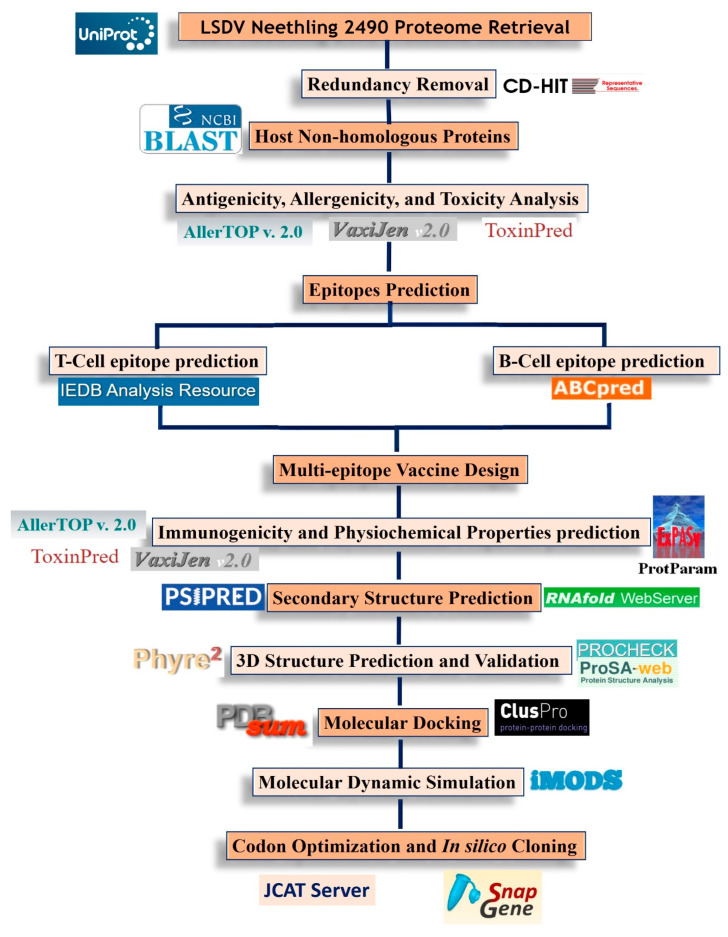
The systematic workflow of the current study.

**Figure 2 bioengineering-10-00430-f002:**
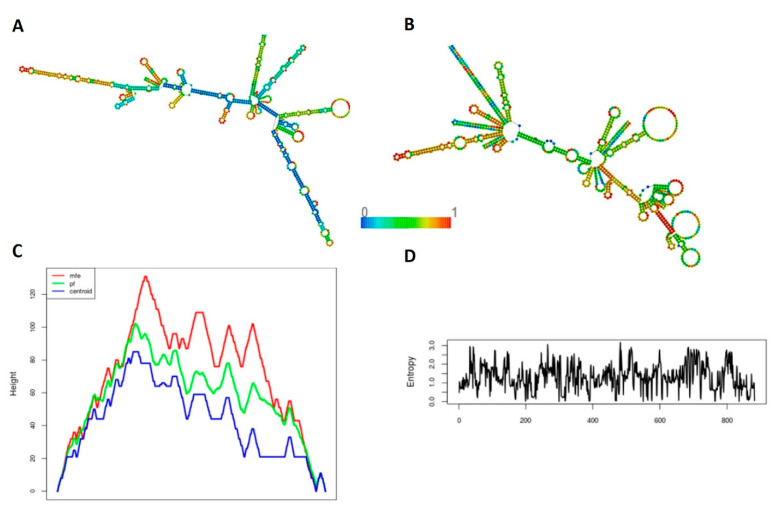
Secondary Structure prediction of the top prioritized LSDV-V2 mRNA construct. (**A**) Model presentation of the minimum free energy of the LSDV-V2 mRNA structure (**B**) The proposed model of the minimum free energy of secondary centroid LSDV-V2 mRNA structure (**C**) The mountain plot representation of the MFE structure, the thermodynamic ensemble of mRNA structure, and the centroid structure. (**D**) The positional entropy plot for each position.

**Figure 3 bioengineering-10-00430-f003:**
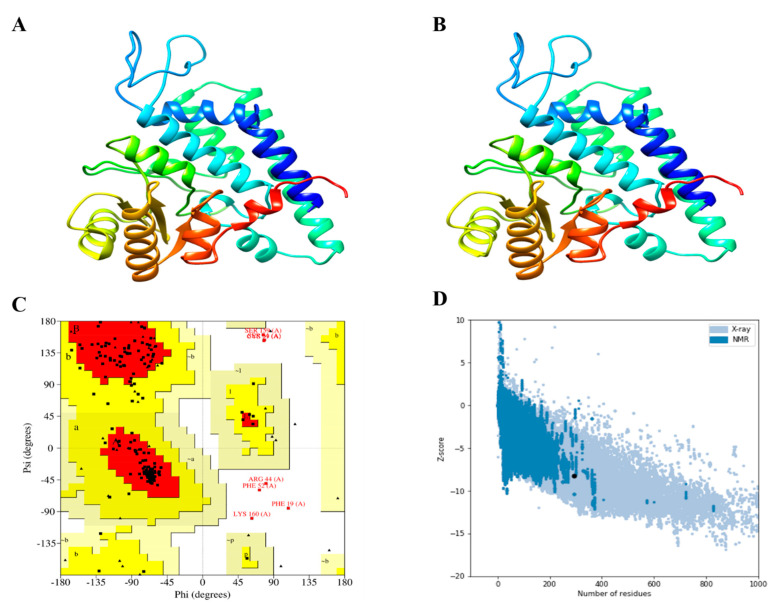
3D structure modeling, refinement, and quality validation analysis of LSDV-V2. (**A**) The 3D model of predicted vaccine construct (**B**) Refine 3D structure of LSDV-V2 (**C**) The Ramachandran plot analysis of LSDV-V2 shows that 88.2% of the residues are present in the favored region, 8.8% were found in the allowed region, while 2.9% were in the disallowed region of the plot. (**D**) ProSA-web plot for LSDV-V2 model.

**Figure 4 bioengineering-10-00430-f004:**
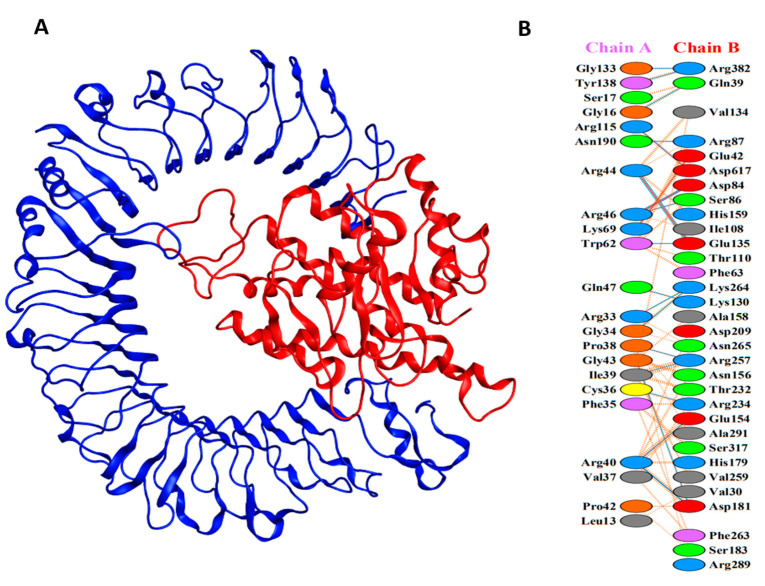
(**A**) The docked complex of LSDV-V2. TLR4 immune receptor is shown in blue while red represents the LSDV-V2 vaccine construct. (**B**) Interacting residues between TLR4 (chain A) and vaccine construct (chain B).

**Figure 5 bioengineering-10-00430-f005:**
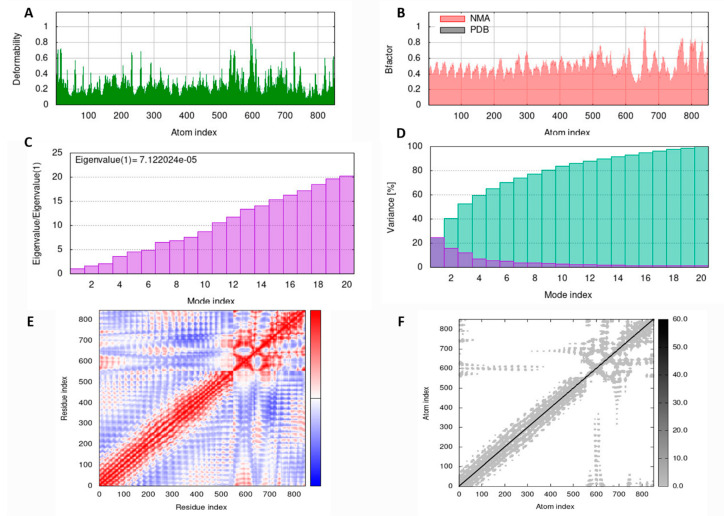
MD simulation results of LSDV-V2 and TLR4 complex obtained by iMODS server. (**A**) deformability (**B**) NMA mobility (**C**) eigenvalues (**D**) purple in the bar shows the individual variances and green depicts cumulative variances (**E**) Covariance map specifies correlated (red), uncorrelated (white), and anti-correlated (blue) motions of paired residues and (**F**) The elastic network model (grey regions indicate stiffer regions).

**Figure 6 bioengineering-10-00430-f006:**
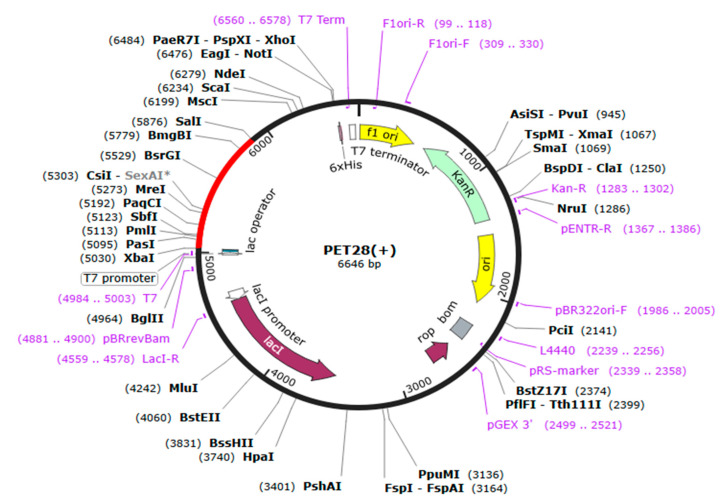
In silico cloning of LSDV-V2 vaccine into pET28a (+) *E. coli* expression vector.

**Table 1 bioengineering-10-00430-t001:** The top-ranked vaccine candidate proteins of LSDV for designing multi-epitope vaccine constructs models.

S/No	Protein GenBank-IDs	Allergenicity AllerTOP2.0	VaxiJen 2.0 > 0.4 Threshold	ToxinPred
1	NP_150447.1	Non-allergen	0.473	Non-toxin
2	NP_150557.1	Non-allergen	0.596	Non-toxin
3	AAK84969.1	Non-allergen	0.473	Non-toxin

**Table 2 bioengineering-10-00430-t002:** The prioritized B-cell epitopes and their immunogenic potential.

Protein IDs	B-Cell Epitopes	ABCPred Score	Antigenicity Score	Allergenicity AllerTop2.0	ToxinPred
NP_150447.1	EGVYLCSITTDTRCNP	0.93	0.6686	Non-allergen	Non-toxin
	NSVIGTNYELLCINTK	0.83	1.5331	Non-allergen	Non-toxin
	SITTDTRCNPKNLALK	0.82	1.1784	Non-allergen	Non-toxin
NP_150557.1	YGLVKKKNNIWVDVNS	0.80	0.8027	Non-allergen	Non-toxin
	LSNIKKSSKGDINACY	0.70	0.5839	Non-allergen	Non-toxin
	SCNYVSYIICVKRLYN	0.68	0.6507	Non-allergen	Non-toxin
AAK84969.1	YTTQQYCNVSPFINDN	0.89	0.5361	Non-allergen	Non-toxin
	KGCIVEFGSQEKVCVT	0.82	0.5036	Non-allergen	Non-toxin
	SFPKDIKLTSNDFNSN	0.74	0.6487	Non-allergen	Non-toxin

**Table 3 bioengineering-10-00430-t003:** The top prioritized MHC1 and MHCII T-cell epitopes.

Protein IDs	MHC1-T-Cell Epitopes	IC50 Value	Antigenicity Score	MHCII Epitopes	IC_50_ Value	Antigenicity Score	Allergenicity AllerTOP2.0	ToxinPred
NP_150447	NVLDYDRSK	1.097615	0.4097	ALIIKEVKRKYL	8.85	0.5982	Non-allergen	Non-toxin
	NSTIALGKN	2.639064	0.6360	LIIKEVKRKYLS	9.05	0.5862	Non-allergen	Non-toxin
	TVNFLNSTI	3.753617	0.7420	IALIIKEVKRKY	12.95	0.9813	Non-allergen	Non-toxin
NP_1505570	AIFMLVSTI	3.269	0.4443	NVSIRHLKVISL	39.5	1.5742	Non-allergen	Non-toxin
	NVSCNYVSY	3.345	1.4980	VSYIICVKRLYN	31.55	0.5581	Non-allergen	Non-toxin
	NYVSYIICV	3.867	0.6540	SIRHLKVISLTY	36.61	1.5129	Non-allergen	Non-toxin
AAK84969.1	DKKGCIVEF	2.357491	0.6565	KTDLSLLKRRIQ	28.47	0.9108	Non-allergen	Non-toxin
	DFWIKFISI	2.84741	0.9557	DLSLLKRRIQKV	28.48	0.5208	Non-allergen	Non-toxin
	NTDDFWIKF	2.988478	0.7614	VFIKRQDVNTVL	45.84	0.5375	Non-allergen	Non-toxin

**Table 4 bioengineering-10-00430-t004:** Physiochemical properties of multi-epitope mRNA vaccine constructs.

Vaccine Constructs	No. of Amino Acids	Molecular Weight	Theoretical PI	Aliphatic Index	Grand Average of Hydropathicity	Instability of Index	GC Content	CAI
LSDV-V1	294	30.96	9.67	77.11	−0.200	20.68 Stable	51.36	1.0
LSDV-V2	294	30.75	9.68	78.78	−0.120	17.85 Stable	49.77	1.0
LSDV-V3	294	30.98	9.64	79.08	−0.061	15.60 Stable	50.68	1.0

**Table 5 bioengineering-10-00430-t005:** The secondary and tertiary structure prediction and structural validation of the vaccine constructs.

Vaccine Constructs	α-Helix	Extended Strand	β-Turns	Random Coils
LSDV-V1	39.12%	21.43%	8.16%	31.29%
LSDV-V2	41.16%	23.13%	8.84%	26.87%
LSDV-V3	34.69%	27.55%	8.16%	29.59%

## Data Availability

All of the relevant data are provided in the form of regular figures, tables, and Appendix A.

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
