# Peer review of "Annotation of Potential Vaccine Targets and Designing of mRNA-Based Multi-Epitope Vaccine against Lumpy Skin Disease Virus via Reverse Vaccinology and Agent-Based Modeling"

_bioengineering, 2023, doi:10.3390/bioengineering10040430_

Round 1

Reviewer 1 Report

1. The manuscript method was used to test the allergenicity, antigenicity and toxicity of the viral protein sequence and epitope sequence respectively. Is it necessary to test the viral protein? Proteins may be toxic or allergenic, but the epitopes in them do not necessarily have such characteristics.

2. For toxicity prediction, the algorithm of polypeptide and intact protein is different. Please use ToxinPred2 for prediction of intact protein.

3.T cell epitope prediction part of the data set missing MHC molecular alleles. Even if the default is used, list it for reference.

4. After the author's sequence screening, only three useful sequences were obtained, whether the proportion was too small compared with the 2490 sequences initially included in the group?

5. Does the article lack population coverage analysis? To characterize the protective effect of vaccines in populations with different MHC gene subtypes.

6. Please list the parameters of molecular docking such as docking pocket information.

Reviewer 2 Report

   In this manuscript, Sehrish et al. investigated an efficient and thermostable multi-epitope-based novel mRNA vaccine model by employing immunoinformatics and reverse vaccinology approaches. The lead B and T-cell epitopes were anticipated from the shortlisted proteins and utilized in the designing of multi-epitopes vaccine constructs. The lead construct model was found to hold all the desirable characteristics to possible elicit a robust immune response with non-allergenicity feature. Molecular docking and MD simulation ensures the strong binding affinity of proposed vaccine construct immune receptor protein. The topic of this manuscript is of interest for the field; however, some data are not convincing, and additional experiments should be conducted to support their conclusions.

   1. In Figure 3, Robetta server was used to create the 3D structure of the designed chimeric vaccine constructs (Fig. 3A) and refined using Galaxy Refine server (Fig. 3B). It is surprising that the 3D model of predicted vaccine construct is exactly the same using the two different methods. If possible, please double check the 3D model constructs.

  2. In this suty, the subtractive genomics and reverse vaccinology approaches were employed to design a novel mRNA vaccine against LSDV, and predicted a highly immunogenic vaccine with a potential to elicit strong immune responses in the bovine host immune system. All these data were obtained by software simulation calculation. Whether the mRNA vaccine was synthesized according to the predicted results, and then its solubility and immunogenicity were analyzed. Or discuss the practicability of this prediction and the success of clinical trials of mRNA vaccine against other viruses that were predicted by this method.

Reviewer 3 Report

Review of "Annotation of potential vaccine targets and designing of mRNA-based multi-epitope vaccine against lumpy skin disease virus via reverse vaccinology and agent-based modeling"

Perhaps the most important aspect of developing a subunit viral vaccine is the need to identify one or more "protective antigens".  These protective antigens need to elicit immune responses (T and/or B cells) that will provide resistance to either infection or disease to the vaccinated host.  The authors of this manuscript fail to identify any "protective antigens" here.  They very elegantly make computer model predictions about what constitute promising candidates for B and T-cell epitopes (which I have no problem with) but are any of these promising candidates going to elicit a protective immune response?  Infections stimulate an array of B and T-cell responses but in most cases, only a specific handful of those responses actually provide protection.  To the authors credit, they do provide one brief sentence at the end indicating that further studies are needed to validate their findings.  It might be worth determining if anyone has identified one or more monoclonal antibodies that have neutralizing effects on LSDV.  At least then there would be a protein(s) on LSDV that you could focus your computer programs on.  The predicted epitopes might have a better chance of providing some protection.  

Further points;

The existing modified-live vaccines are effective (11 commonly used modified-live vaccines available) and are considered the primary strategy for controlling LSDV outbreaks.  They do suffer from causing mild disease in vaccinated animals and can recombine with virulent field strains infrequently.  Over-attenuated live vaccine strains do not stimulate strong protective immunity and make poor vaccine candidates.  Some inactivated vaccines have been used and do provide protection but require a boost.  Your mRNA vaccine approach would most likely require a boost as well.  

There is a definite need for a DIVA vaccine strategy for LSDV and your subunit vaccine approach could be useful for this to distinguish between vaccinated cattle and cattle infected with field strains by incorporating a unique peptide sequence (B-cell epitope) that could be tested by ELISA for peptide antibodies as a measure of vaccination.  

Your RNA vaccine would likely need to be stored at -80 degrees C, thawed and used that same day.  This could be a logistical problem for countries with inadequate infrastructure.  

Minor issue;

I found it confusing in section 2.10 (lines 191-192) where you discuss "codon adaptation to acquire sophisticated expression of the designated vaccine in the E. coli strain K12 expression system".  This makes it sound like you are expressing the protein in E. coli when in fact you are using the pET28 vector for the transcription of your mRNA for expression in bovine cells.  I would suggest substituting the term transcription for expression system and making it clearer that the codon adaptation is for expression in bovine cells (not E. coli).  
